# Classification for *Penicillium expansum* Spoilage and Defect in Apples by Electronic Nose Combined with Chemometrics

**DOI:** 10.3390/s20072130

**Published:** 2020-04-09

**Authors:** Zhiming Guo, Chuang Guo, Quansheng Chen, Qin Ouyang, Jiyong Shi, Hesham R. El-Seedi, Xiaobo Zou

**Affiliations:** 1School of Food and Biological Engineering, Jiangsu University, Zhenjiang 212013, China; 2Division of Pharmacognosy, Department of Medicinal Chemistry, Uppsala University, Box 574, SE-75 123 Uppsala, Sweden

**Keywords:** gas sensors, electronic nose, apple, pattern recognition, variable selection

## Abstract

It is crucial for the efficacy of the apple storage to apply methods like electronic nose systems for detection and prediction of spoilage or infection by *Penicillium expansum*. Based on the acquisition of electronic nose signals, selected sensitive feature sensors of spoilage apple and all sensors were analyzed and compared by the recognition effect. Principal component analysis (PCA), principle component analysis-discriminant analysis (PCA-DA), linear discriminant analysis (LDA), partial least squares discriminate analysis (PLS-DA) and K-nearest neighbor (KNN) were used to establish the classification model of apple with different degrees of corruption. PCA-DA has the best prediction, the accuracy of training set and prediction set was 100% and 97.22%, respectively. synergy interval (SI), genetic algorithm (GA) and competitive adaptive reweighted sampling (CARS) are three selection methods used to accurately and quickly extract appropriate feature variables, while constructing a PLS model to predict plaque area. Among them, the PLS model with unique variables was optimized by CARS method, and the best prediction result of the area of the rotten apple was obtained. The best results are as follows: R_c_ = 0.953, root mean square error of calibration (RMSEC) = 1.28, R_p_ = 0.972, root mean square error of prediction (RMSEP) = 1.01. The results demonstrated that the electronic nose has a potential application in the classification of rotten apples and the quantitative detection of spoilage area.

## 1. Introduction

Fruits and vegetables occupy a vital position in the modern agricultural economy and are an important part of agriculture. Its development can promote the adjustment of agricultural structure and promote the development of society and economy. Due to its high nutritional value and its richness in minerals and vitamins, apples have become a very popular agricultural product around the world. According to Food and Agriculture Organization of the United Nations (FAO)statistics, apple’s total world production reached 86.14 million tons in 2018 [1]. Despite this fact, apples are prone to spoilage during transportation and storage. Apples are more susceptible to a variety of internal and external factors leading to infections of various diseases and pathogenic microorganisms, resulting in serious postharvest losses. At present, the monitoring mode of temperature, humidity and carbon dioxide in the storage of fruits and vegetables is unable to realize the monitoring and early warning of corruption. At the same time, there is a huge demand for product testing, and the severe challenge of quality and safety control warrants the investment of a rapid and nondestructive detection method for the corrupted apple, which is of a great significance to promote the development of the apple industry.

For a long time, conventional detection methods such as gas chromatography [2] and gas chromatography-mass spectrometry [3] have been applied to the detection of gases in fruits and vegetables, but these instruments are expensive and time-consuming. In recent years, research in this area has used machine vision [4] and near-infrared spectroscopy [5] however, these methods were not able to characterize the bacteria. At present, electronic nose technology has made a great contribution to the safety and quality detection of fruits and vegetables, showing its potential in the detection field. The electronic nose is generally composed of a sample processing system, several chemical sensor arrays and a pattern recognition system [6]. The electronic nose obtains the characteristic information through a variety of sensors, and processes it with the appropriate pattern recognition analysis method, which can realize the qualitative and quantitative aspects of the spoilage [7]. The electronic nose detection has the advantages of being fast, simple and non-destructive [8]. Ying [9] and Xu [10] respectively used e-nose to evaluate the freshness of litchi and measured the storage time of litchi under three different conditions. Dai [11] combined electronic nose with headspace solid phase microextraction (HS-SPME)-GC-MS to analyze the volatile components in the fermentation process of Tremella aurantialba. Torri [12] and Sun [13] used electronic noses to predict the freshness of pineapple and fish, respectively.

In recent studies, electronic nose has also been applied to the identification of fruit and vegetable varieties and pathogenic fungal. Adak [14] and Güney [15], obtained four kinds of aroma data of strawberry, lemon, cherry and melon by the electronic nose, and successfully classified the aroma data and distinguished three kinds of fish, horse mackerel, anchovy and whiting. Pan et al. [16] used a combination of electronic nose and GC-MS to detect and classify postharvest pathogenic fungal diseases of strawberry fruits, and the accuracy rate of discriminating the types of strawberry fruit fungal infection was 96.6%. Bonah used electronic nose to classify and identify bacterial foodborne pathogens [17] and summarized the methods and pattern recognition tools used by electronic nose in the detection of foodborne pathogens [18]. Biondi et al. [19] used the electronic nose to detect potato ring rot and brown rot, where the linear discriminant analysis (LDA) used passive sampling with an accuracy of 81.3%. However, most previous studies have focused on the detection of aromas of fruits and vegetables and the determination of water, sugar and soluble solids in fruits and vegetables. The use of electronic nose technology for fruit and vegetable spoilage gas research has been seldom used, and the quantitative study of fruit and vegetable spoilage has not been reported. The composition and proportion of fruits and vegetables are time-varying and complex, which is different from the industrial production. The constitute and content of gases produced in the process of fungal infection of fruits and vegetables are affected by the internal components and the external environment, which puts forward high requirements for the sensitive detection of gas sensors.

Based on previous studies, the purpose of this study includes the following three aspects: (1) using electronic nose to obtain the gas information of apple before and after inoculation with *Penicillium expansum* and to study the response law of electronic nose to apples with different degrees of spoilage, (2) different pattern recognition methods are used to classify apples with different levels of corruption, and feature variables are selected through different variable selection methods to establish a PLS prediction model of apple corruption area, (3) research on apples with different levels of corruption through electronic noses, with a view to provide reference and solutions for early warning of apple corruption. The general procedures of electronic nose are illustrated in Figure 1.

## 2. Materials and Methods

### 2.1. Sample Preparation

The Fuji Apple was collected from the supermarket of Zhenjiang City as the experimental sample. Apples with basically a similar size and maturity were selected and randomly divided into 4 groups (30 sample sets per group). The maturity, size and color of the selected apple are similar, and there is no obvious mechanical damage on the surface. After the samples were shipped to the laboratory, the samples were randomly divided into 4 groups of 30 each. 

The inoculated mold is *Penicillium expansum* purchased from China Center of Industrial Culture Collection (CICC). The fungal strains were grown on potato glucose agar (PDA) at 27 °C and 85% relative humidity for one week. After the fungal spores were separated from the PDA surface, the concentration was measured with a blood cell counter and adjusted to a final concentration of 1 × 10^5^ spores mL^−1^ with sterile saline solution (0.85% (V/V) NaCl).

Apple samples were washed with 75% alcohol and air-dried on a sterile workbench. Then, use the inoculator to drill three holes (3 mm diameter and 5 mm depth) at different locations on each apple surface. The suspension of *Penicillium expansum* was inoculated into the hole through the inoculation ring, and the sterile membrane was used to cover the equator of the apple. Next, place the apple in a constant temperature and humidity box at 25 °C and 60% humidity. When the apple began to rot, it was divided into three grades according to the surface area of the plaque (the radius was 0.5–1 cm, 1.0–1.5 cm and 1.5–2.0 cm, respectively). Thirty samples were taken from the blank group and samples of different grades for the electronic nose test.

### 2.2. Electronic Nose Sampling

In this experiment, a Airsense PEN3 portable electronic nose is used, which is mainly composed of sensor array for gas collection. The sensor array is composed of 10 sensors, and the characteristics are shown in Table 1.

First, apple samples were taken from the Humidity Chamber. Then, 30 samples were placed in a 1000 mL beaker, covered with two layers of fresh-keeping film and placed at 25 °C for 1 hour for an electronic nose test. Electronic nose empty run once, purify electronic nose sensor. The test conditions were as follows: cleaning time 180 s, waiting time 5 s and test time 160 s. Repeat the above steps 4 times to obtain the electronic nose information of blank samples and corrupted samples, respectively.

### 2.3. Pattern Recognition Methods

#### 2.3.1. Principal Component Analysis

Principal component analysis (PCA) is a common algorithm in dimension reduction, and it can also be regarded as a multivariate statistical analysis method to master the main contradictions of things [20]. It became the most commonly used feature extraction method. It is a method of data dimensionality reduction by using linear mapping, and at the same time, it can remove the correlation of data, to keep the variance information of the original data to the maximum extent. Therefore, PCA can be used to process the original data of electronic nose, eliminate noise and unimportant features, extract feature information, greatly simplify the difficulty and complexity of problem processing and improve the data processing speed and the anti-interference ability of the original data.

#### 2.3.2. Linear Discriminant Analysis

Linear discriminant analysis (LDA) is the best projection direction obtained by the method of finding the extreme value of Fisher criterion function so that after the sample is projected in this direction, it enables the largest inter-class dispersion and the smallest intra-class dispersion [21]. It is a classical algorithm for pattern recognition. The main idea is to project the classified samples onto a one-dimensional line to minimize the variance within the category and maximize the variance between categories [22]. Compared with PCA, LDA can increase the distribution and mutual distance within the same category, collect information from all data and improve the classification accuracy.

#### 2.3.3. K-Nearest Neighbor

K-nearest neighbor (KNN) method is also called Reference Sample Plot Method. The KNN algorithm is a simple, effective and non-parametric classification method [23]. The core idea of KNN is simply to give a prediction target, then calculate the distance or similarity between the prediction target and all samples, then select the first K samples that are closest to each other and then use these samples to vote for decisions [24]. The basic steps are as follows: The training set and test set are constructed, and then the k value is set. The distances between the samples of the training set and the test set were arranged in descending order, and k samples with small distances were selected as the k nearest neighbors of the test samples to obtain the class with the largest number of k nearest neighbors.

#### 2.3.4. Principal Component Analysis Discriminant Analysis

Principal component analysis-discriminant analysis (PCA-DA) is a PCA-based analysis method [25]. To combine the two algorithms of PCA and LDA, principal component analysis was first performed on sample data to achieve the purpose of dimensionality reduction. At the same time, a corresponding model is established to classify unknown samples. The specific steps are as follows: map the sample data to the feature subspace with PCA algorithm, select the number of PCA principal components according to the contribution rate or other indicators and finally use LDA algorithm to carry out linear classification in the subspace. PCA-DA takes into account the class member information provided by the auxiliary matrix in the form of code when constructing factors, so it has an efficient discrimination ability, and at the same time improves the validity and validity of the model.

#### 2.3.5. Partial Least Square Discriminant Analysis

Partial least square discriminant analysis (PLS-DA) algorithm is a discriminant analysis method based on PLS regression. The basic idea is to establish a qualitative analysis model according to the characteristics of the known sample set [26]. It trains the characteristics of different processing samples (such as observation samples and control samples) respectively, generates training sets and tests the credibility of training sets [27]. In the process of analysis, the overlapping part of numerous chemical information can be eliminated to extract the data most relevant to the sample category, that is, to maximize the difference between the extracted data of different categories, making the analysis data more accurate and reliable and improving the accuracy of classification discrimination.

### 2.4. Multivariate Calibration Methods

#### 2.4.1. Partial Least Square

Partial least squares (PLS) is a multivariate statistical analysis method, which has been widely used in fields such as fruits and vegetables. It can better summarize the information of independent variables and better explain dependent variables by extracting effective comprehensive variables. PLS can overcome the problem of multivariate collinearity caused by the interaction between independent variables, thus greatly eliminating interference factors and improving the accuracy of prediction [28]. It cannot only overcome the collinearity problem, but also emphasize the interpretation and prediction function of independent variables to dependent variables when selecting the feature vectors, eliminate the influence of regression noise and make the model contain the least number of variables.

#### 2.4.2. Synergy Interval Partial Least Square

Synergy interval partial least square (SI-PLS) is an extension of the interval PLS wavelength selection method proposed by Norgard [29]. The main idea is to use a part of the sub-interval of the full region to build a local PLS model. This part of the spectrum is most sensitive to changes in the target variable while avoiding the influence of other unrelated interval information and noise. The specific implementation steps are briefly described as follows: based on interval PLS, the sub-intervals of several local models with higher accuracy in the same interval division are combined, and the RMSEC value is the joint model measurement index, selected from all models the best joint sub-interval (RMSEC is the smallest). The PLS model based on the best joint subinterval has the strongest predictive ability [30]. The electronic nose data has a large amount of information, and it is easy to introduce too much useless data during the modeling process, which reduces the prediction accuracy of the model, and the data interval needs to be filtered. Therefore, SI-PLS can divide the data into several intervals and select feature intervals. Modeling can reduce the amount of modeling data, simplify the model and improve robustness. 

#### 2.4.3. Genetic Algorithm

Genetic algorithm (GA) is a kind of algorithm that imitates the survival of the fittest and the survival of the fittest rules by referring to the evolutionary mechanism of species competitive selection in the process of biological evolution in nature [31]. Different from the general intelligent algorithm, the genetic algorithm searches from multiple solutions and evaluates multiple values at the same time to avoid falling into the local optimal solution. Based on the fitness function, the genetic algorithm implements the iterative optimization of the individual structure reorganization in the population by imposing genetic operations such as selection, crossover, variation, etc. on the individuals in the population [32]. Through the combination of GA algorithm and PLS algorithm, the feature information can be extracted better and a highly accurate and adaptive model can be established.

#### 2.4.4. Competitive Adaptive Reweighted Sampling

Competitive adaptive weighted sampling (CARS) is a method based on Darwin’s “survival of the fittest” theory to select characteristic variables from the redundant variables. Through the adaptive reweighted sampling technique, the optimal subset of variables is selected, which improves the prediction ability of the model and reduces the prediction variance. To a certain extent, this algorithm can overcome the combination explosion problem in the variable selection, select the optimized subset of variables, improve the prediction the ability of the model and reduce the prediction variance [33]. At the same time, the algorithm introduces the exponential decay function, so it controls the retention rate of variables, thus has high calculation efficiency, and is suitable for variable selection of high-dimensional data.

## 3. Results and Analysis

### 3.1. Analysis of Apple Gas Components

The response curves of 10 sensors in a single rotten apple electronic nose are shown in Figure 2a. Different color curves correspond to the response signals of different sensors at corresponding time. Before sample collection, the response value of each sensor (G/G0 or G0/G) is 1. Take the name of each sensor as the abscissa, and the average value of each sensor’s response signal to different degrees of corruption as the ordinate to make Figure 2b. The intensity of the gas response signal of each sensor to the rotten apple was expressed. It can be seen from the figure that the sensors R7, R2, R6, R9 have strong response signals. Combined with the response characteristics of each sensor in Table 1.

According to the sensitivity of the sensor to different compounds, R7 sensor is sensitive to sulfur-containing organics, R2 sensor is sensitive to nitrogen oxides, R6 sensor is sensitive to a wide range of methane, R9 sensor is sensitive to aromatics and organic sulfides, which shows that the main components in the gas of rotten apple are the above substances. The contribution rates of R2, R6, R7 and R9 were 15.25%, 16.61%, 65.36% and 2.7%, respectively, with a cumulative contribution rate of 99.92%. Therefore, four characteristic sensors (R2, R6, R7 and R9) were selected for subsequent analysis and compared with the analysis results of the selected sensors.

### 3.2. Principal Component Analysis

The results of the PCA using the data onto all sensors and feature sensors are shown in Figure 3. The contribution rate of PC1 and PC2 in Figure 3a is 87.86% and 10.95%, respectively, and the contribution rate of cumulative variance in the first two principal components is 98.81%; the contribution rate of PC1 in Figure 3b is 89.24%, the contribution rate of PC2 is 10.22%, and the contribution rate of cumulative variance of the first two principal components is 94.46%. As can be seen from the figure, the principal component analysis method can well classify the blank group with the third-level apple. However, the samples of the first and second grades partially overlap. Therefore, it is possible to classify apples with different degrees of corruption by other pattern recognition methods.

### 3.3. Classification Models of Spoilage

#### 3.3.1. LDA Model

LDA results using all sensor and feature sensor data are shown in Figure 4. Figure 4a shows that the accuracy of the correction set is 98.61%, and the accuracy of the prediction set is 95.83%, both of which exceed 95%. Figure 4b shows that the accuracy of the correction set and the prediction set are both 95.83%. Therefore, the LDA algorithm has good performance in the classification of apples with different levels of corruption, and the results of classification using the data of all sensors are slightly better than the results onto the data of feature sensors.

#### 3.3.2. KNN Model

The results of the analysis of KNN algorithm using all the data of all sensors, are shown in Figure 5a,b. The classification effect is ideal, the accuracy of the training set and prediction set are 98.61% and 95.83%, respectively. The results of KNN algorithm analysis using the data of feature sensors are shown in Figure 5c,d, and the accuracy of the training set and prediction set are 93.06% and 100%, respectively.

#### 3.3.3. PCA-DA Model

The PCA-DA algorithm was used to establish discriminant models of apples with different levels of corruption (a total of 72 samples), and the prediction set (a total of 48 samples) were used as external verification. The specific results of using all sensor data are shown in Appendix A. The accuracy of the training set and the prediction set was 95.83%. Similarly, the results of using the feature sensors data analysis are shown in Appendix A. The accuracy of the training set and prediction set are 97.22% and 100%, respectively.

#### 3.3.4. PLS-DA Model

The classification results of apple samples by PLS-DA algorithm using all sensor data are shown in Appendix A. From the table, the accuracy of training set is 100%, while that of prediction set is only 93.75%. Appendix A shows the classification results of apple samples by PLS-DA algorithm using feature sensors data. The accuracy of the training set and the prediction set was 100%.

#### 3.3.5. Compare Different Classification Algorithms

The results of classifying apples with different degrees of corruption by using data onto all sensors and feature sensors using four algorithms are shown in Table 2. Comparing the results of using all sensor data to classify each algorithm, it can be concluded that the accuracy of PCA-DA is the lowest (both the training and prediction sets are 95.83%), and PLS-DA has the highest accuracy (100% in training set and 93.75% in prediction set). In the same way, compared with accuracy of classification using feature sensors, the accuracy of LDA is the lowest, the accuracy of training set is 95.83%, and the accuracy of prediction set is 95.83%. The accuracy of PCA-DA classification is the highest, the accuracy of prediction set and training set was 97.22 and 100%, respectively.

Therefore, by comparing the classification results of all sensors and feature sensors, it is found that the accuracy of classification by PCA-DA algorithm is the highest. By selecting the data of the gas sensitive feature sensors of rotten apple for classification and analysis, the irrelevant data is greatly reduced, and the classification accuracy is improved, which shows that the electronic nose can be used to classify rotten apple caused by inoculation of *Penicillium expansum* with different degrees.

### 3.4. Quantitative Models of Apple Spoilage Area 

#### 3.4.1. PLS Models

Partial least squares (PLS) can effectively solve the problems of multiple data, correlation and overlap. Figure 6a,b shows the patch area distribution of the PLS model using all the sensor and feature sensor data, respectively. Table 3 contains the results of the PLS model. Among them, the effect of feature sensor data modeling is better, the results are R_c_ = 0.919, RMSEC = 1.66, R_p_ = 0.945, RMSEP = 1.40. Therefore, PLS model based on all data does not achieve the expected prediction accuracy, because it contains a lot of irrelevant information, which reduces the prediction ability of the model.

#### 3.4.2. SI-PLS Models

Si divides the data interval involved in the modeling into several different sub-intervals, and then combines three or four sub-intervals to build a PLS model, which can greatly reduce irrelevant information and obtain the best prediction model. Among them, the best result of modeling with all sensor data are to decompose all data into 24 sub intervals, among which the effective intervals are the 7th, 9th and 18th, which are used for apple spoilage area prediction, as shown in Figure 6c. The optimal results of Si-PLS model based on electronic nose data are shown in Table 3. The results of apple spoilage area were R_c_ = 0.929, RMSEC = 1.56, R_p_ = 0.938, RMSEP = 1.46. Similarly, Figure 6d is the best result of the apple spoilage area model when the feature sensor data is divided into 20 sub intervals and the effective intervals are the 6th, 13th and 19th. The results of apple spoilage area were R_c_ = 0.938, RMSEC = 1.45, R_p_ = 0.954, RMSEP = 1.27.

#### 3.4.3. GA-PLS Models

GA-PLS is an excellent optimization algorithm by combining the global optimization search capability of GA algorithm with the multicollinearity problem solving capability of PLS algorithm. Figure 6e,f shows the apple spoilage area distribution of GA-PLS model using electronic nose data, and the predicted results of GA-PLS are shown in Table 3. Among them, the results of modeling using all sensor data are: R_c_ = 0.917, RMSEC =1.65; R_p_ = 0.925, RMSEP = 1.61, slightly lower than the results of modeling using feature sensors. Therefore, the best results are: R_c_ and RMSEC are 0.939 and 1.45, and the best R_p_ and RMSEP are 0.948 and 1.36, respectively.

#### 3.4.4. CARS-PLS Models

CARS method regards each variable as an individual, and selects the optimal variable by adaptive weighted sampling technology. At the same time, this algorithm can control the retention rate of variables, showing high computational efficiency, and is suitable for variable selection of high-dimensional data. Table 3 describes the best CARS-PLS model. Figure 6g,h displays the apple spoilage area distribution of the GARS-PLS model using all sensor data. The results of the best PLS model based on the car analysis of all sensor data are R_c_ = 0.937, RMSEC = 1.48; R_p_ = 0.941, RMSEP = 1.45, slightly lower than that based on the feature sensors data (R_c_ = 0.953, RMSEC = 1.28; R_p_ = 0.972, RMSEP = 1.01).

#### 3.4.5. Comparisons of Different PLS Models.

Using the three variable selection methods of SI, GA and CARS to analyze the data of all sensors and characteristic sensors, and the results of the PLS model for the apple spoilage area of the putrid apple were shown in Table 3. The correlation coefficient of each model is greater than 0.8, which shows that the model has good prediction performance. At the same time, it also shows that the electronic nose can be used to determine the apple spoilage area with different degrees of corruption. Through the analysis of the results of each model, it is found that all the results of modeling with feature sensors data are better than those of modeling with all sensor data. This is because the electronic nose has a total of ten sensors, and each sensor has its own sensitive gas. At the same time, the amount of collected data is relatively large and complicated and contains a lot of irrelevant information. Therefore, selecting feature data that is more sensitive to the corrupt apple gas can filter out a lot of irrelevant information, improving the accuracy of classification judgment and the stability of the model.

Through the CARS algorithm, the data is filtered to remove a lot of irrelevant information, which improves the efficiency and accuracy of the model. The model correlation coefficients R_c_ and R_p_ for predicting apple spoilage area of spoiled apples was increased to 0.953 and 0.972, respectively. It shows that it is feasible to quantitatively detect the apple spoilage area of spoiled apples using an electronic nose.

## 4. Conclusions

In this study, the gas information of an apple before and after inoculation with *Penicillium expanse* was collected by electronic nose to classify the apples with different rotten degrees and predict the apple spoilage area. By analyzing the sensitivity of each sensor to the rotten apple gas and the contribution rate of each sensor to the results, it is concluded that sensors 2, 6, 7 and 9 are feature sensors. In this study, five pattern recognition algorithms (PCA, LDA, KNN, PCA-DA and PLS-DA) were used to classify apple samples, and four algorithms (PLS, SI-PLS, GA-PLS and CARS-PLS) were used to predict the Apple spoilage area of apples with different degrees of decay. The results of the analysis using the information of all sensors are compared with the results of the analysis using the information on feature sensors, and the latter is found to have a higher accuracy. Among them, the classification accuracy of PLS-DA for different levels of corruption reached 100%. The PLS model with unique variables optimized by CARS algorithm obtained the best prediction result of the Apple spoilage area. The best results are as follows: R_c_ = 0.953, RMSEC = 1.28, R_p_ = 0.972, RMSEP = 1.01. According to the accuracy of each model, the electronic nose can effectively detect and identify apples with different degrees of corruption. At the same time, CARS algorithm is used to select feature variables to simplify the analysis process and improve the analysis accuracy and model stability. The CARS-PLS model can predict the degree of apple corruption and has tremendous potential for practical applications, which is beneficial to the quality monitoring of apple during storage.

## Figures and Tables

**Figure 1 sensors-20-02130-f001:**
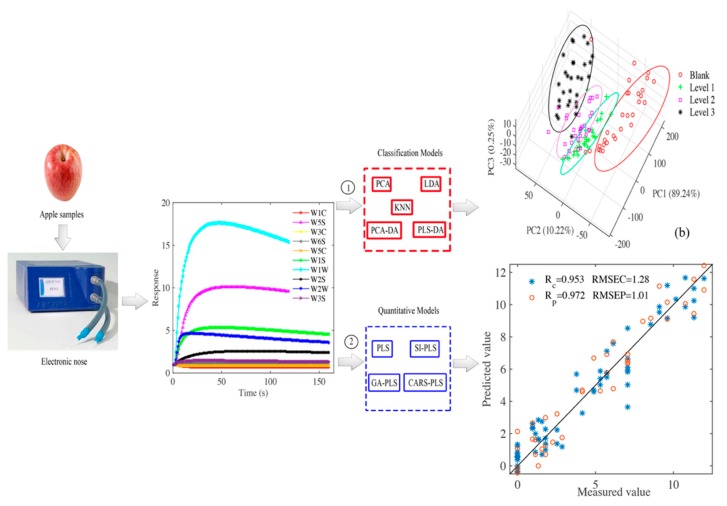
Schematic procedures of classification and prediction of defects of *Penicillium expansum* in apples by electronic nose combined with chemometrics.

**Figure 2 sensors-20-02130-f002:**
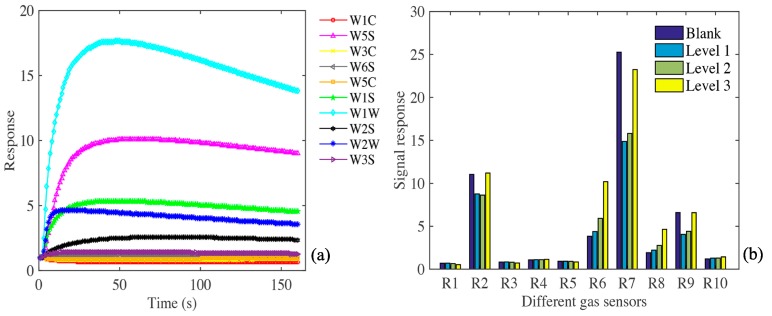
(**a**) Data of each sensor of a single corrupt apple; (**b**) response signals of various sensors to apple gases with different degrees of corruption.

**Figure 3 sensors-20-02130-f003:**
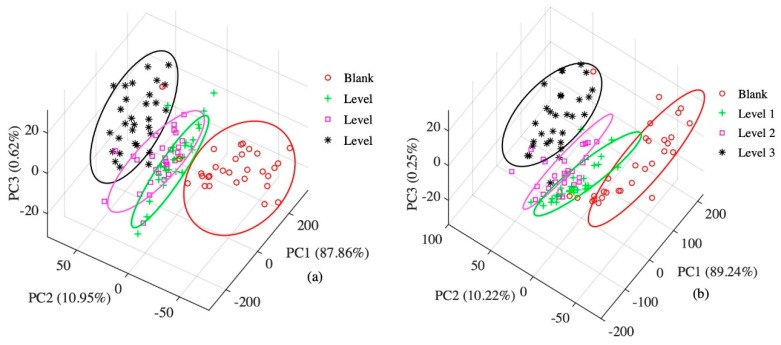
(**a**) Principle component analysis (PCA) results using data from all sensors; (**b**) feature sensors.

**Figure 4 sensors-20-02130-f004:**
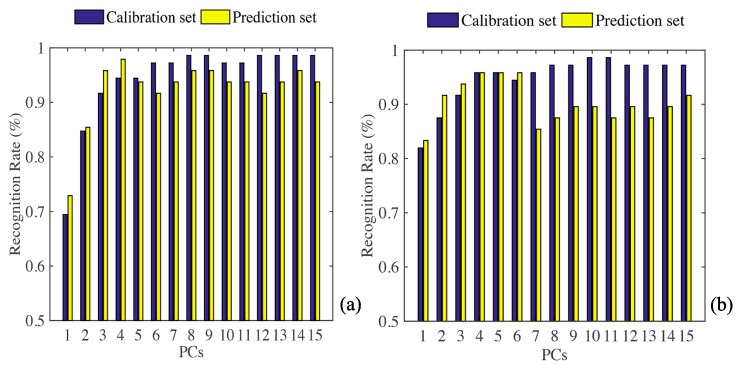
(**a**) Linear discriminant analysis (LDA) results using data from all sensors; (**b**) feature sensors.

**Figure 5 sensors-20-02130-f005:**
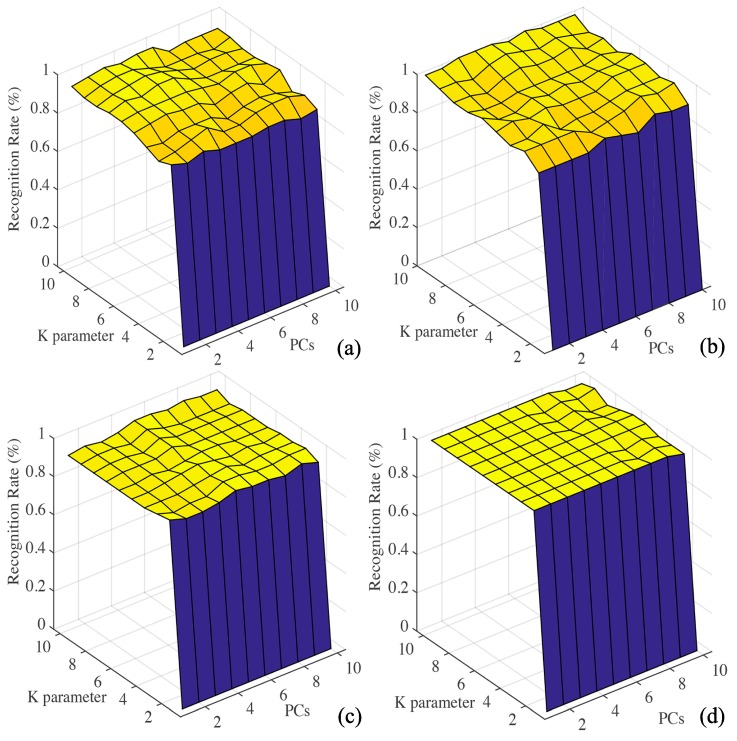
(**a**,**b**) K-nearest neighbor (KNN) results using data from all sensors; (**c**,**d**) feature sensors.

**Figure 6 sensors-20-02130-f006:**
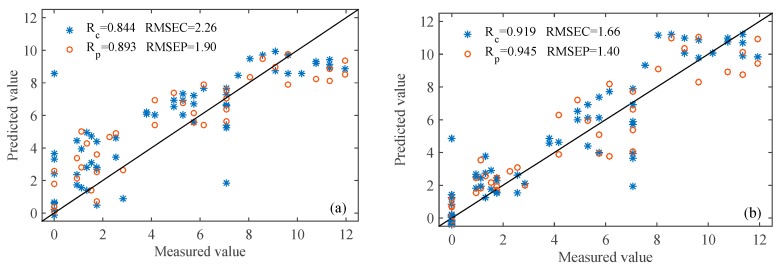
(**a**,**b**) Quantitative model results for all sensor versus characteristic sensor information using PLS; (**c**,**d**) Si-PLS; (**e**,**f**) GA-PLS; (**g**,**h**) CARS-PLS.

**Table 1 sensors-20-02130-t001:** Response features of the gas sensor array.

Number in Array	Sensor	Main Attribute	Typical Target
R1	W1C	Aromatic compounds	C_6_H_5_CH_3_
R2	W5S	Nitrogen oxides	NO_2_
R3	W3C	Ammonia and aromatic molecules	C_6_H_6_
R4	W6S	Hydrogen	H_2_
R5	W5C	Alkanes, aromatic compounds	C_3_H_8_
R6	W1S	Broad methane	CH_4_
R7	W1W	Sulfur-containing organics	H_2_S
R8	W2S	Broad alcohols	C_2_H_5_OH
R9	W2W	Aromatics, organic sulfides	H_2_S
R10	W3S	Methane and aliphatics	CH_4_

**Table 2 sensors-20-02130-t002:** Classification results using data from all sensors and feature sensors.

Algorithm	All Sensors (R1-R10)		Feature Sensors (R2 R6 R7 R9)	
	Calibration Set	Prediction Set	Calibration Set	Prediction Set
LDA	98.61%	95.83% 5.83%	95.83%	95.83%
KNN	98.61%	95.83%	95.83%	100%
PCA-DA	95.83%	95.83%	97.22%	100%
PLS-DA	100%	93.75%	95.83%	100%

**Table 3 sensors-20-02130-t003:** Results of an optimal PLS model using different data based on different variable selection methods.

Model	All Sensors (R1-R10)				Feature Sensors (R2 R6 R7 R9)			
	Calibration Set		Prediction Set		Calibration Set		Prediction Set	
	R_c_	RMSEC	R_P_	RMSEP	R_c_	RMSEC	R_P_	RMSEP
PLS	0.844	2.26	0.893	1.90	0.919	1.66	0.945	1.40
SI-PLS	0.929	1.56	0.938	1.46	0.938	1.45	0.954	1.27
GA-PLS	0.917	1.65	0.925	1.61	0.939	1.44	0.942	1.42
CARS-PLS LSPLS	0.937	1.48	0.941	1.45	0.953	1.28	0.972	1.01

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
