# Peer review of "Classification for Penicillium expansum Spoilage and Defect in Apples by Electronic Nose Combined with Chemometrics"

_sensors, 2020, doi:10.3390/s20072130_

Round 1

Reviewer 1 Report

The article entitled: Classification for Penicillium expansum spoilage and defect in apple by electronic nose combined with chemometrics.

It presents a beautiful study especially from the point of view of analyzing the data obtained. In fact, the sample analyzed and the sensors used are not particularly new. The data analysis approach is very nice.

I would suggest suggesting adding a part relating to the choice of sensors on the PEN3.
- How were the sensors chosen?
- Have they been selected on the basis of the matrix to be analyzed?
- What are the VOCs that vary between a contaminated and non-contaminated sample? Having knowledge of which compounds can be formed by contamination, you can choose more selective and sensitive sensors

Author Response

Thank you for your recognition of our work and valuable comments. These comments have certain reference value for the revision and improvement of our paper, and have important guiding significance for our research.

We fully agree with your observations and have made changes in the above statement. The selection of feature sensors is in the section 3.1 results and analysis (P:10, L: 236), and the interpretation is modified in the revised draft. " The contribution rates of R2, R6, R7 and R9 were 15.25%, 16.61%, 65.36% and 2.7%, respectively, with a cumulative contribution rate of 99.92%. Therefore, four characteristic sensors (R2, R6, R7 and R9) were selected for subsequent analysis and compared with the analysis results of the selected sensors." (P:11, L: 253-256).

Thanks again for your suggestions, all of which are very important and have important guiding significance for my future research work.

Reviewer 2 Report

The paper showed an application of the electronic nose with chemometrics used for detection and prediction of Penicillium expansum spoilage or infection of apples. The result is very good. Generally, it is very difficult to find spoilage samples in these experiments. The author used inoculated apples as spoilage samples tactfully. And the result also exhibited CARS can be a good feature extraction method for PLS. The experiment design and result is very useful. The paper suitable for the Sensors.

  1. There is a typing error in line 82, “Apple s” should be “Apples”
  2. The minor scale of Axis of PCs of The fig 5(a) should be as same as fig.5 (b、c、d)
  3. There is also a typing error in the last row of Table 3, “CARS-” should be “CARS-PLS”

Author Response

Thank you for your recognition of our work and valuable comments. These comments have certain reference value for the revision and improvement of our paper, and have important guiding significance for our research.

We fully agree with your observations and have made changes in the above statement. The "Apples" has been changed to "Apples"(P:4, L:96). The scale problem in Figure 5 has been unified (P:13, L:288), and the CARS-PLS in Table 3 have been added (P:18, L:383).

Thank you again for your suggestions, all of which are very important and have important guiding significance for my future research work.

Reviewer 3 Report

In this paper,the spoilage of apples has been studied using  a commercial e-nose using a variety of statistical methods. These type of works were novel in the nineties. The methodoogy is adequate, reslts are good and the conclusions are sound but this is not reserach, it is just a comprobation that a commercial electronic nose works 

The novelty of the work is very low. These type of works were carried out in the nineties. The only point to be mentioned is the number of statistical methods used, but this is not new neither. 

Author Response

Thank you for your insightful review. These comments have certain reference value for the revision and improvement of our paper, and have important guiding significance for our research. In recent years, electronic nose technology has been widely used in the quality inspection of fruits and vegetables, such as the determination of water, sugar and maturity.

In the process of fruit and vegetable harvesting, some surface damages are easily recognized by human or machine vision and processed accordingly. But for the hidden damage of fruits and vegetables, especially the spoilage caused by microorganisms, it is difficult to be accurately identified. It is easy to increase the spoilage in the storage process, resulting in a large area of microbial infection, which brings huge economic losses to farmers.

This study is based on the previous research, using electronic nose to classify different degrees of corruption apple, and combining with chemometrics to predict the area of corruption. We have supplemented the introduction part of this article. “However, most previous studies have focused on the detection of aromas of fruits and vegetables and the determination of water, sugar and soluble solids in fruits and vegetables. The use of electronic nose technology for fruit and vegetable spoilage gas research has been seldom used, and the quantitative study of fruit and vegetable spoilage has not been conducted.” (P:3, L:75-79). We believe this will make the novelty of this article clear.

Thank you again for your suggestions, all of which are very important and have important guiding significance for my future research work.

Round 2

Reviewer 3 Report

As I stated before, in my opinion the paper has a complete lack of novelty. A commercial electronic nose is used to analyze apples. I am not sure wheter the problem of the infection by Penicillium is interesting for the food industry. But, the sensors and statistics used are conventional and not new at all. For this reason, I beleive that this paper is not interesting for a journal dedicated to sensors.

Author Response

Thank you for your constructive comments. As an important part of sensors, gas sensors play an important role in research and industrial application. More and more researches have carried out, and practical applications are extensive. We agree with the reviewer’s comments. The sensors and algorithms used in this paper are relatively conventional. Our research group has also carried out profound research in this area nearly 20 years, and also proposed or improved some new algorithms, such as decision tree classification algorithm, ant colony optimization algorithm and simulated annealing algorithm. This research is based on the demand of anti-corrosion and anti-mildew in fruit and vegetable industry. At present, the monitoring mode of temperature, humidity and carbon dioxide in the storage of fruits and vegetables is unable to realize the monitoring and early warning of corruption. According to the literature retrieval, the application of electronic nose for monitoring and early warning of fruit and vegetable spoilage fungi has not been reported, which shows that this paper is innovative in solving key problems in the food industry. In the revised manuscript, we added the introduction of the industry background (Page 2, Line 40-43). Non-destructive monitoring of food quality and safety using gas sensors has become one of the hot research spots, which may be the reason why “non-destructive Sensors and Machine Learning for Food Safety & Quality Inspection” is a special issue of Sensors. As an important food source, the amount of fruits and vegetables in storage and circulation is huge, meanwhile, the economic loss caused by the postharvest corruption of fruits and vegetables is also huge. The key to controlled atmosphere storage of fruits and vegetables is to maintain quality and prevent spoilage and mold. In this paper, gas sensor array was used to rapidly identify the corrupt degree caused by dominant apple spoilage fungus, and the scale of the corruption was quantitatively predicted. Meanwhile, the principal component analysis method was used to screen and evaluate the sensors, so as to simplify the sensor array and achieve effective prediction, which is valuable for the practical application of electronic nose. In the revised manuscript, we added the following sentences to descript the special requirements for fruit and vegetable corruption monitoring (Page 3, Line 81-85). “The composition and proportion of fruits and vegetables are time-varying and complex, which is different from the industrial production. The constitute and content of gases produced in the process of fungal infection of fruits and vegetables are affected by the internal components and the external environment, which puts forward high requirements for the sensitive detection of gas sensors.” In addition, we modified the conclusion section to make the paper clearer (Page 20, Line 403-404, Line 417). Thank you again for those thoughtful comments.
